# Crashworthiness Analysis of Thin-Walled Square Columns with a Hole Trigger

**DOI:** 10.3390/ma16114196

**Published:** 2023-06-05

**Authors:** Michał Rogala, Jakub Gajewski

**Affiliations:** Department of Machine Design and Mechatronics, Faculty of Mechanical Engineering, Lublin University of Technology, 20-618 Lublin, Poland; j.gajewski@pollub.pl

**Keywords:** thin-walled structure, crashworthiness, trigger, finite element method

## Abstract

Thin-walled structures dynamically loaded with an axial force are the subject of this study. The structures work as passive energy absorbers by progressive harmonic crushing. The absorbers were made of AA-6063-T6 aluminum alloy and subjected to both numerical and experimental tests. Experimental tests were performed on an INSTRON 9350 HES bench, while numerical analyses were performed using Abaqus software. The energy absorbers tested had crush initiators in the form of drilled holes. The variable parameters were the number of holes and their diameter. The holes were located in a line 30 mm away from the base. This study shows a significant effect of the hole diameter on the values of the stroke efficiency indicator and mean crushing force.

## 1. Introduction

Due to increasingly restrictive regulations on passive safety in motor vehicles, since the 1980s, vehicle producers have begun to improve their design. In the history of automobiles, initially, due to the relatively small number of cars and road incidents such as crashes and accidents, designers focused on developing a vehicle whose strength would be high and that would give the impression of a very rigid body. Given the speeds reached by the first cars, high structural rigidity was not problematic. Over time, designers recognized the problem of overload occurring during collisions, which in extreme cases, could prove fatal to passengers. In the early 1990s, scientific publications began to appear addressing the issue of passive safety [1,2], which intensified the dynamics of the development of this branch of science. Initially, rigid structural elements began to take a different form. Elements began to be used, which on impact, were to deform plastically by which they absorbed mechanical energy to protect passengers. These structures were initially located in the front section and called crash boxes. Over the years, attention began to turn to the possibility of energy absorption by other bodily components, such as the floor, side pillars, and doors. Passive structures are based on loss of stability and buckling at a precise location [3,4,5,6,7,8]. As a result of scientific research, it was determined that energy absorbers of a certain width and wall thickness were able to ripple in an orderly (harmonic) manner [9,10]. The first crash boxes were based on an aluminum/steel column with a simple crush initiator designed to start the creasing process at a precise location [11,12]. Such disruption of the column structure, also known as a trigger, is widely used. Its use avoids global buckling of the structure, which results in the loss of the column’s potential to absorb a significant amount of mechanical energy. In addition, thanks to the use of a trigger, the initiation of the first plastic hinge occurs in a way that reduces the efficiency of the first fold, but however, significantly reduces the maximum force, the value of which translates into the amount of overload generated. Crush initiators most often take the form of a geometric disturbance in the form of embossing or discontinuities in the material. A trigger in the form of ribbing, thanks to the continuity of the column structure, reduces the risk of material tearing at high beater speeds. Examples of poles with this type of trigger [13,14,15] have shown that it reduces the magnitude of the first force peak while maintaining the energy absorption potential. Another interesting approach in the way of modeling overstressing was presented by Ferdynus et al. [16], who presented the side edge crumpling of an aluminum column with a square cross-section. Their study showed that the depth of embossing significantly affects the magnitude of the maximum crushing force generated at the start of crushing, with little effect on the average crushing force. This is very important when modeling the mechanism of crush initiation, so as not to reduce the total energy output of the column. Another way of triggering the crush of thin-walled structures is through cavity processing, i.e., cutting holes of different shapes on the sidewalls, and cutting material on the edges of the columns [17,18,19,20]. The holes on the columns perform the same role as the embossing, i.e., changing the stiffness of the column and allowing the initiation of a local plastic joint. The performance of this type of initiator is also high; however, it depends on its geometric parameters. Based on research published by Rogala et al. [21], it was found that for a hexagonal initiator, the dominant parameter is the width of the cutout. The height of the hole improves the performance of the initiator to a much lesser extent, while it may cause unstable behavior of the energy absorber during non-axial impact. The next important aspect is the position of the initiator relative to the base of the column. The amount of energy absorbed is mainly related to the achieved buckling posture, and this changes with any interference with the geometric parameters of the trigger. Studies have shown that the size of the plastic hinge depends on the width of the profile [9,22]. The use of numerical methods, especially the finite element method, is particularly useful in research [23,24,25]. It allows for the reduction in the time of the conducted research as well as the reduction of the research group in experimental bench tests [26,27,28,29].

In analyzing the field of passive energy absorbers while studying the role of the trigger, which weakens the structure of the column but is an essential element for the proper crushing of the structure, researchers began to use structural reinforcement. The first methods of reinforcing crash boxes were multi-cell profiles [30,31,32,33], involving filling the column with additional thin walls inside. The development of technology for the manufacture of multi-cell structures has brought the honeycomb structure, which by its orderly structure, has also found applications in passive energy absorbers [34,35,36]. Due to the fact that foam absorbs energy by deforming continuously, they have also become popular in scientific research. Great interest is being shown in the use of metal foams, especially aluminum. According to the technological parameters used, aluminum foam can reach a density up to ten times lower than that of solid material while retaining great potential as an absorber of mechanical energy [37,38,39].

Nowadays, due to the rapid development of manufacturing technology, e.g., metal powder sintering, metal casting, and composite structures have found applications as energy absorbers. Composite metal–ceramic foam based on aluminum and silicon carbide (SiC) particles is an interesting approach in the study of absorption of mechanical energy. Dynamic tests of axially crushed thin-walled columns filled with different lengths of the porous structure have been presented [40]. In addition, in one study, the authors compared the energy-absorbing properties of a composite foam with traditional aluminum foam of similar density. The study showed a higher rate of absorbed energy due to the greater ductility range of pure aluminum while emphasizing that both foams exhibit good energy-absorbing properties [41,42,43,44].

Due to the wide range of applications and the possibility of changing the mechanical properties of structures made of fiber laminates, studies are increasingly describing the use of composite structures as passive energy absorbers. The studies presented by various authors have been characterized by different layouts and fiber materials; however, the mechanism of destruction was often based on the creation of a dedicated matrix that enhanced the failure of the composite structure [45,46]. As these materials lack a plastic range, energy absorption is therefore based on material damage.

## 2. Crashworthiness Efficiency Indicator

In order to correctly analyze the crush of passive energy absorbers over the years, indicators began to appear in scientific publications to determine the performance of thin-walled structures. Based on the graph of the force-shortening relationship (Figure 1) and the basic parameters recorded during crushing, such as the total shortening of the sample or the maximum force, it is possible to determine crush efficiency indicators. The first basic quantity needed is the energy absorbed during the crush, described by Equation (1); however, in scientific research to compare the geometric changes introduced, the comparison of models is performed with the same initial energy.
(1)$$EA(dx)=\int_{0}^{dx}F\left(x\right)dx$$

Another quantity necessary for further analysis of energy absorbers is the mean crushing force (*MCF*), which is defined as the quotient of the energy absorbed and the total crushing distance (2).
(2)$$MCF=\frac{EA(dx)}{dx}$$

The next indicator, crush load efficiency (*CLE*), shows the relationship between the peak crushing force (*PCF*) and the *MCF*:(3)$$CLE=\frac{MCF}{PCF}\times 100\%$$

Stroke efficiency (*SE*) is a ratio comparing the value of the shortening of the column to its initial length (4). This is the intermediate ratio needed to determine the total capacity of the column described by Equation (5): (4)$$SE=\frac{U}{L_{o}}$$

Total efficiency (*TE*) represents both the value of the course of the crushing force and compares it with the relative shortening that occurred during the test. On the basis of one indicator, we were able to approximate its energy absorption capacity (crashworthiness):(5)TE=CLE×SE

## 3. Research Methodology

This study used the finite element method (Abaqus) as well as experimental models to verify the numerical one. The test specimen took the shape of a square column with a crush initiator in the form of drilled holes located at a height of 30 mm from the bottom edge of the absorber. The fixed parameters were a height of 200 mm, a cross-section of 40 × 40 mm, and a side wall thickness of 1.2 mm. The columns were made of AA-6063-T6 aluminum alloy, the model of which was determined by static tensile testing. The test specimens had an initiator in the form of one, three, or five holes, aligned as shown in Figure 2. In addition, the holes had a variable diameter in the range of 1 to 4 mm. All specimens were tested with the same initial conditions (i.e., a constant mechanical energy of 1700 J, which was the resultant mass of the 70 kg beater and its velocity of 7 m/s).

### 3.1. Materials and Methods

The specimens were made of AA-6063-T6 because of their good mechanical properties, including plastic behavior (Table 1 and Table 2). The properties of the structure were also suitable for subsequent experimental testing on a dynamic crush test stand for thin-walled profiles. Aluminum-6063-T6 is known for its excellent weldability, which makes it a popular choice for constructing frames, railings, and other structural components. Its relatively low weight, good machinability, and high thermal conductivity also make it useful for manufacturing heat sinks, electrical components, and automotive parts. Table 1 shows the mechanical properties provided by the manufacturer. Due to the accuracy of the tests conducted, the numerical analyses were based on our own tests of the mechanical properties shown in Table 2.

The above table shows the chemical composition, temperature treatment as well as basic mechanical properties of the aluminum alloy. Table 1 presents the yield strength limit (Re), tensile strength limit (Rm), percent elongation for a 50 mm long strain gauge (A50), and Brinell hardness.

The material model was based on the elastic domain of the Young’s modulus and Poisson’s ratio and on the plastic domain of the multilinear stress–strain characteristics. According to scientific research, aluminum alloys are not susceptible to strain rate, making it unnecessary to use the Johnson–Cook constitutive material model to obtain valid results [47,48,49,50].

### 3.2. FEM Analysis

Numerical studies were carried out in two stages using the Abaqus CAE software. The first step was to obtain a buckling form, which was then implemented for dynamic analysis. The boundary conditions were simplified to two non-deformable plates serving as a beater and a base according to Figure 2. In the lower base, there is a reference point that collects force values in order to obtain the force-shortening characteristic.

The numerical tests carried out made it possible to present the values of force as a function of displacement, which is the basic graph for the subsequent determination of crush efficiency indicators more extensively described in Section 2. During the analysis, 1000 measurement points were obtained, which made it possible to determine the indicators with a high degree of accuracy.

The size of the finite elements is closely related to the accuracy of the tests carried out using the finite element method. In view of the above, a sensitivity analysis was conducted for a column without a crush initiator. The results of the analysis are shown in Table 3, which includes the finite element size, duration, and discrepancy of shortening relative to the physical test specimen. The results showed a significant increase in analysis duration for a finite element smaller than 2.5 mm. In addition, the columns with finite elements of 2 and 2.5 mm showed the highest convergence. Given the results presented, the numerical analysis used an element size of 2.5 mm for the column and 4 mm for the non-deformable plates serving as the beater and base.

### 3.3. Experimental Analysis

The impact test was conducted on an Instron CEAST 9350HES drop hammer, the general view of which is shown in Figure 3. The stand consists of a test table to which the specimen was attached. The stand has a strain gauge sensor built into the base that reads measurement data such as force and acceleration values. In addition, the dynamic testing hammer has a tup, to which the mass was applied, thus translating into the energy obtained.

The attachment of the sample was carried out with dedicated steel cubes. Their dimensions were matched to the inner sizes of the aluminum profile. The purpose of the grips was to stabilize the sample during the crush and to properly convert the impact of the striker into the crush of the absorber.

The thin-walled models were manufactured using an AA-6063-T6 aluminum alloy. The cross-section size was 40 mm × 40 mm, and the wall thickness was 1.2 mm. The length of the profile was 200 mm. The crush initiator, in the form of drilled holes, was made on a tabletop drilling machine with the configuration shown in Figure 2.

## 4. Results

The results of the tests are presented for a group of numerical models and experimental samples. Tests using the finite element method were carried out for three configurations of holes—that is, one, three, or five, located parallel to the edge of the base. The distance was constant for all tested models and equal to 30 mm. In addition, this study determined the effect of the hole diameter on the crushing process of thin-walled profiles in the range of 1–4 mm in 0.5 mm intervals. The results of the conducted tests are presented in the form of plots of force as a function of shortening, in order to observe the occurrence of the force peaks (Figure 4, Figure 5 and Figure 6). In addition, crush efficiency indicators were determined and are presented in column plots. The indices analyzed in this section are described in more detail in Section 2.

The first group of analyzed cases was a passive absorber with one axially located crush initiation hole. The hole was located 30 mm from the edge of the column base and was on two opposite walls. Figure 4 shows the force-shortening characteristics for one initiation hole of different diameters. Because the weakening of the structure due to the discontinuity of the column wall was too small, the results with one hole did not cause a significant change in the characteristics. The occurrence of force peaks was similar for all curves in Figure 4. The case that stands out is the column with a single hole with a diameter of 4 mm. Despite the small difference between the tested hole diameters, the impact is evident compared to the smooth column without an initiator.

The initiation of the crushing process in the form of material discontinuity obtained by drilling holes is evident for the three and five hole models, as seen in Figure 5 and Figure 6, respectively. The effect of three holes initiating the crush is visible in the initial force peak as well as in the total shortening of the samples. The largest force peak (PCF) is characterized by models with the smallest holes—that is, 1 mm and 1.5 mm. These models had a high PCF, but the duration of the peak was small and, thus, translated into little accompanying overload. Small hole diameters resulted in mechanical energy being absorbed over a small distance, resulting in a higher energy-absorbing potential of the passive absorber. For larger hole diameters, a reduction in the force peak at the beginning of the crush was noticeable, as well as a difference in the shortening of the specimens. Studies show that the use of drilled holes in the right configuration makes it possible to initiate the crush and control the crush depending on the expected effects. Passive absorbers are used in special mechanical constructions in this regard, and the diversity of their application is a great advantage.

The last group of tested initiators consisted of five holes placed in a row according to Figure 2b. The plots of the force as a function of shortening presented in Figure 6 show the greatest differences depending on the diameter of the holes used. The PCF values ranged from 42 to 47 kN, and the effect of diameter is much more visible than with one or three holes of the same diameter (Figure 4 and Figure 5). For the models with 1 mm and 1.5 mm diameter initiators, similar behavior to that of the three-hole initiators is noticeable. The samples generated the largest initial force peak as seen in Figure 6. The consequence of a large overload was a significant decrease in force and low efficiency of the first fold. The force peaks occurring later were offset from the others in terms of the force-shortening characteristic. All samples were loaded with a constant energy of 1,700 J; however, the sample with the smallest hole diameter, absorbing energy along the largest path. This would indicate a relatively low mean crushing force (MCF) and low crushing force efficiency (CLE). In order to correctly analyze the discussed trigger cases, it was necessary to compare the results for different diameters. The selected crush efficiency indicators are presented in the form of column charts (Figure 7 and Figure 8).

The curves shown in Figure 4, Figure 5 and Figure 6 map their behavior through crush efficiency factors more broadly described in Section 2. Based on the numerical data, the PCF ratio was determined, which is directly related to the behavior of the structure in the initial phase of crushing. The initial force peak reflects the formation of the first plastic hinge, which, according to the theoretical assumptions described by Jones [2], is dependent on the geometric parameters of the column. Figure 7 shows that by using an appropriate configuration of the crush-initiating holes, the PCF value could be reduced by up to 20% for different hole sizes. The value is much higher when referring to a smooth column without a crush initiator. Such a structure, through much higher stiffness in the direction of impact, generates larger force values for the initiation of the first fold, propagating the further crush of thin-walled passive energy absorbers.

Analyzing the behavior of the PCF index (Figure 7) and TE index (Figure 8), the most obvious relationships were for the case with five initiating holes. In the case of maximum force, the values decreased in an inversely proportional manner to the diameter of the holes. Due to the geometric limitations of the column (i.e., 40 mm in width), the diameter of the holes could not exceed 4 mm. The cases with one and three initiating holes were characterized by non-linear changes for the discussed indicators. These cases were characterized by other properties, which are described in more detail in the paragraphs below (see also Figure 4 and Figure 5). These holes could fulfill their purpose, depending on the expected effects. Passive absorbers, depending on the expected effects, can absorb maximum values along the shortest path or, on the contrary, generate minimum loads, for example, for passengers inside a motor vehicle.

Experimental verification was carried out on the example of the two selected cases characterized by variation in the discussed crush efficiency indicators. The first case was a column with five holes with a diameter of 1 mm, while the second had three holes with a diameter of 4 mm. In both cases studied, the verification was conducted on the example of two specimens labeled S1 (specimen 1) and S2 (specimen 2).

The transposed results show the value of the force as a function of shortening, which presents the location of the formation of the hinges and the values of the occurrence of force peaks. This is due to the type of research carried out, i.e., dynamic analysis, during which progressive harmonic crush occurs. Such studies are subject to certain imperfections due to simplified boundary conditions during numerical simulations or the sampling of the sensor in the experimental study.

This study shows that samples S1 and S2 reproducibly illustrated the behavior of a passive energy absorber. For both samples, the force peaks occurred at the same location on the thin-walled column, which corresponds to the specified shortening of the sample. The values of absorbed mechanical energy by individual plastic hinges correspond to each other. The total shortening of the specimen for all curves in Figure 9 corresponds to the other verified cases. In the experimental study, there was a noticeable tendency for the plastic hinge to split into a force peak, visible on the graph, and a subsequent force spike, which relates to a single peak on the graph for numerical analysis. This effect may have been due to the finite element method. The method simplifies the results used to some extent and the mesh used, and the size of the finite element defines the accuracy of the obtained results. If a very fine element was used, the accuracy of the results could be affected, but the analysis time would be significantly increased and the computational cost would be much higher.

The quality of the verification results for sample D4-3, characterized by three holes with a diameter of 4 mm, was as high as for sample D1-5. The results are also presented in the form of force-shortening characteristics and a graphical presentation of crushed thin-walled samples (Figure 10). The verification of the above sample showed a slightly off-axis crush for sample S1. Due to the slight skewing of the sample, the characteristic shows an increase in the force waveform at the beginning of the crush. The magnitude of the second force peak is larger, indicating that a higher value of mechanical energy was absorbed in this stage. This is related to the shape of the characteristics at the later stage of the crush for the curve defined in green. Despite this behavior of the sample, the shortening value seen in Figure 10 is almost identical for both verified cases, which confirms its good energy-absorbing capabilities, which resulted in the biomechanical properties. Due to the magnitude of the force that was generated, it affects overload that is unsafe for human life during accidents and hence, there is a real correlation between crush-initiating elements and passenger safety. The comparison of numerical results with experimental data allowed us to draw similar conclusions. The convergence of characteristics was at a high level, and the values of absorbed energy at the various stages of crushing refer to each other.

## 5. Conclusions

The results of the presented research are related to the process of the crush initiation of passive energy absorbers. These are mainly used in motor vehicles, and their task is to absorb mechanical energy at the lowest possible overload accompanying the crush. The use of discontinuities as one of the methods of crush initiation is well known; however, the authors have shown in previous studies that horizontal discontinuities located on a column initiate crush in the best way. Based on the results of previous studies, columns with one, three, or five holes located in a line 30 mm from the base were analyzed. Numerical studies verified by the bench experimental tests showed that the use of holes has a positive effect on selected aspects of crush. Observing the results shown in Figure 4, Figure 5 and Figure 6, it can be seen that the number of holes and their diameter affect the shortening of the specimen, which is related to the value of the generated force peaks. Given constant initial conditions (i.e., a mechanical energy of 1700 J for all analyses), the difference in shortening is an indicator of further potential for energy absorption. However, due to the use of energy absorbers as an internal passive safety system—that is, the protection of vehicle occupants—an important aspect is the values of the overload generated during the crush.

Overload is directly related to force values, determined by the PCF parameter, and reduced with the use of an appropriate initiator configuration. In addition, it was observed in the analyses that the crush initiation holes affecting the initial force peak translated into the location of the remaining plastic joints. According to the theory of the formation of plastic hinges, their location depends on the form of buckling; however, with the help of crush initiators, it is possible to properly control the crush to improve crush efficiency indicators. The total efficiency (TE) of the structure, observed in Figure 8, improved significantly, especially for five evenly spaced holes. The extreme values of the diameter of the tested initiator showed an increase in the index from 0.15 to 0.25, or of 66%. The significant improvement in the index is mainly due to the decrease in PCF seen in Figure 7, but the increasing MCF index also contributed to the improvement in overall performance. Through the higher average values of the force waveform, we note the same absorbed energy over a shorter path.

In summary, it can be concluded that the use of a crush initiator in the form of drilled holes is a low-cost, simple-to-implement option for improving indices. Studies have shown improvements in selected indices by up to several tens of percent. However, this method also carries some risks. Using too much discontinuity in the material in the form of drilled holes can result in material cracking. If a wall or edge fracture occurs, the structure is crushed uncontrollably, and its potential is wasted. Thus, when drilling holes, the edges must not have any defects that may indicate the risk of wall damage.

## Figures and Tables

**Figure 1 materials-16-04196-f001:**
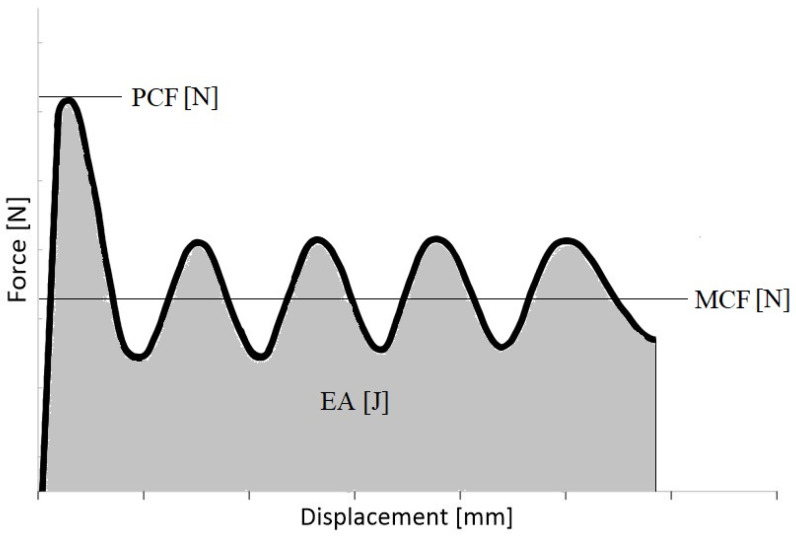
Exemplary force-shortening diagram.

**Figure 2 materials-16-04196-f002:**
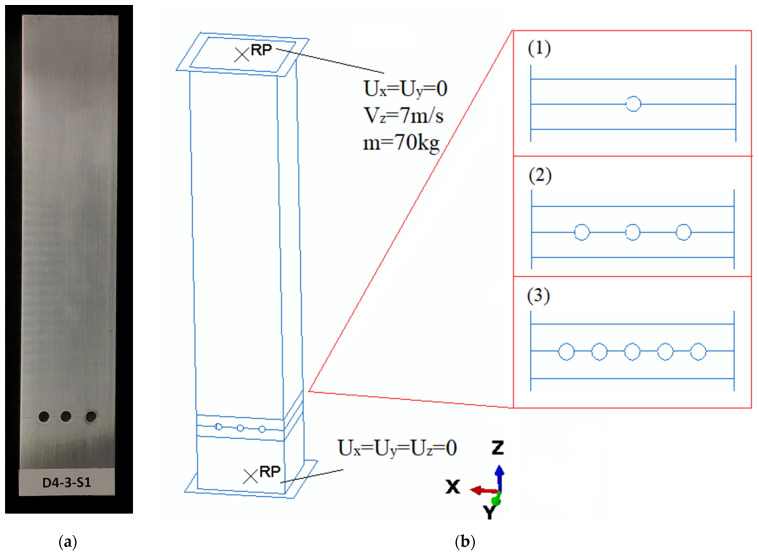
General views of the specimen: (**a**) physical front view and (**b**) the model with three different trigger configurations where (1–3) presents a schematic appearance of the initiator.

**Figure 3 materials-16-04196-f003:**
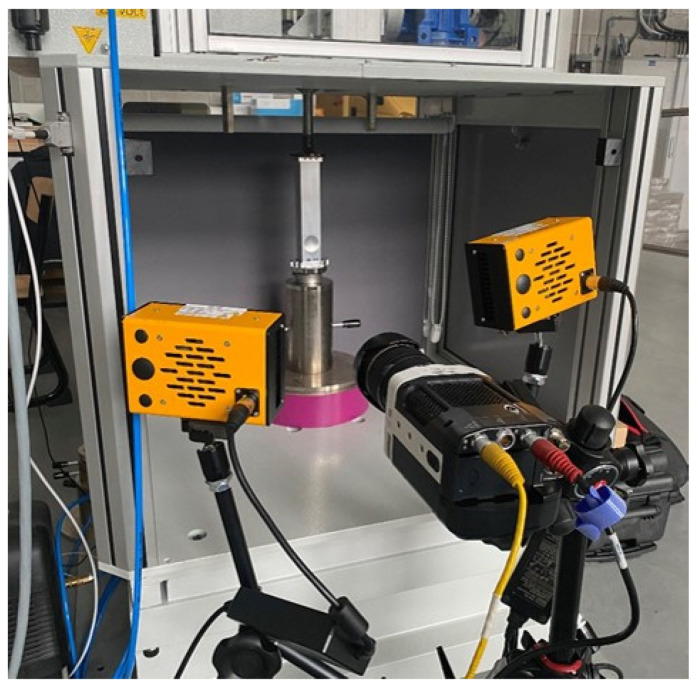
General view of the experiment station.

**Figure 4 materials-16-04196-f004:**
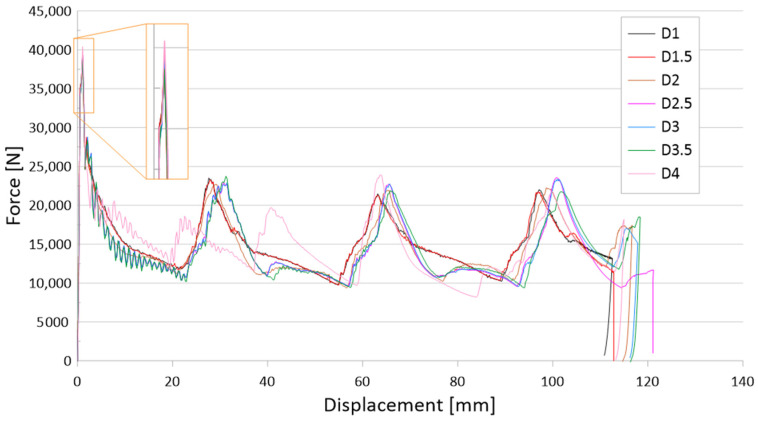
Force-shortening diagram of thin-walled structures with a one-hole crush initiator.

**Figure 5 materials-16-04196-f005:**
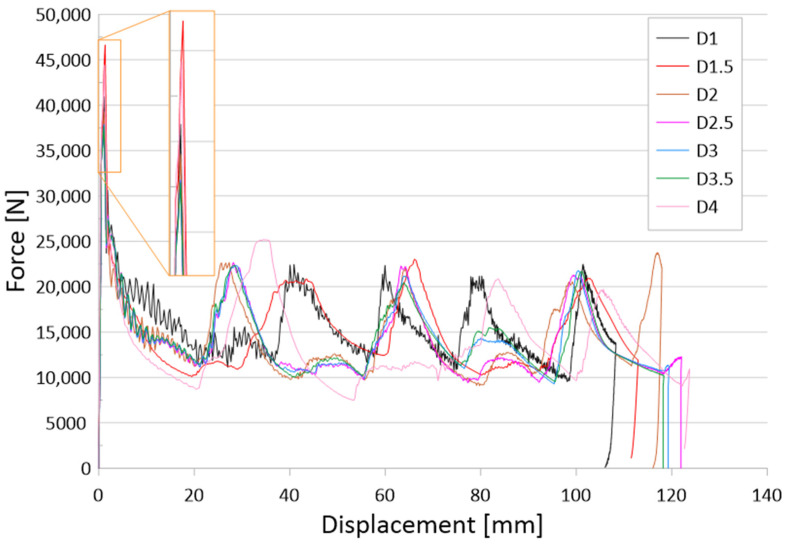
Force-shortening diagram of thin-walled structures with a three-hole crush initiator.

**Figure 6 materials-16-04196-f006:**
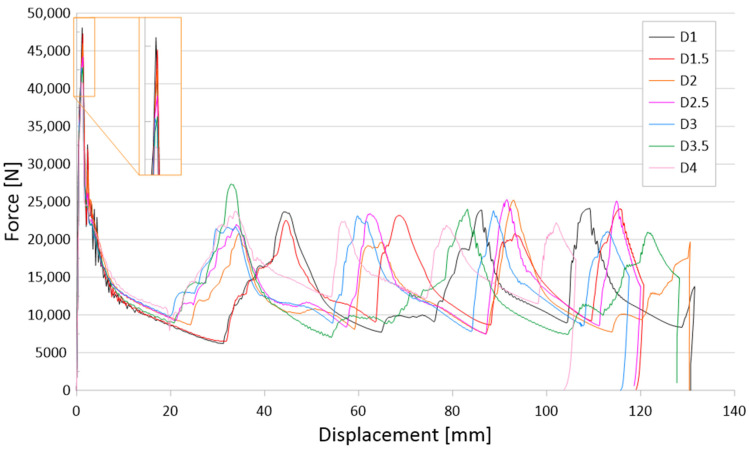
Force-shortening diagram of thin-walled structures with a five-hole crush initiator.

**Figure 7 materials-16-04196-f007:**
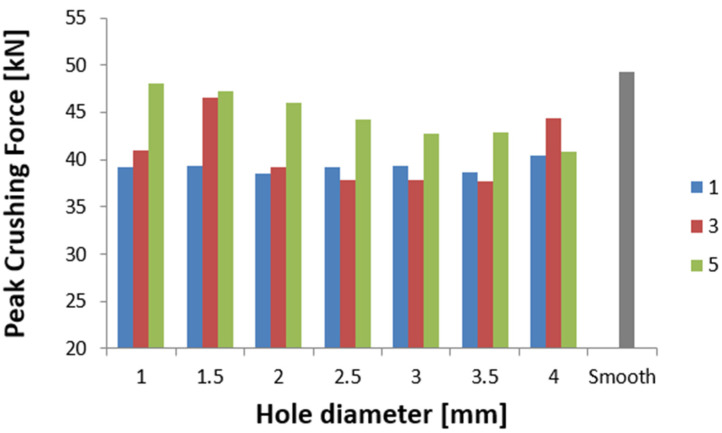
Effect of the diameter and number of holes on the PCF indicator.

**Figure 8 materials-16-04196-f008:**
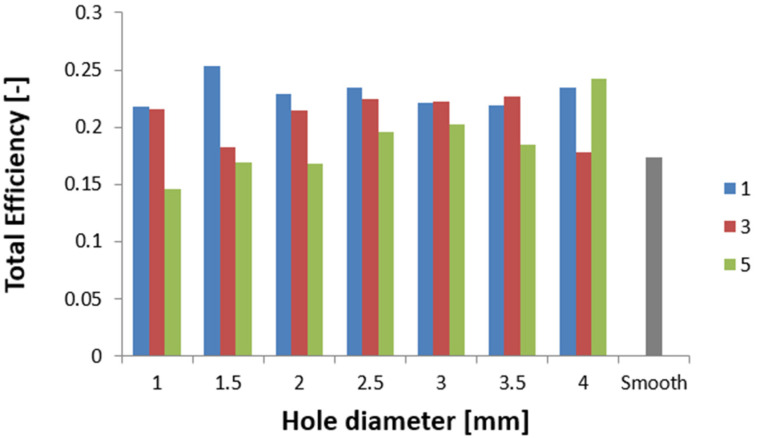
Effect of the diameter and number of holes on the TE indicator.

**Figure 9 materials-16-04196-f009:**
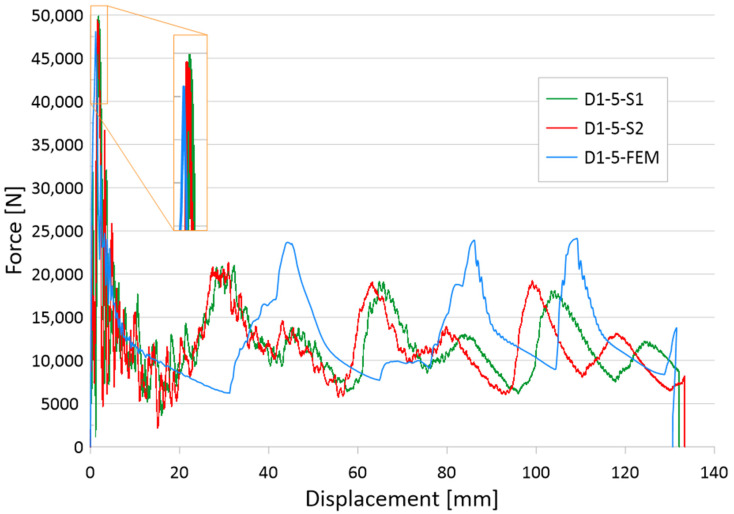
Comparison of numerical and experimental analysis based on D1-5 models.

**Figure 10 materials-16-04196-f010:**
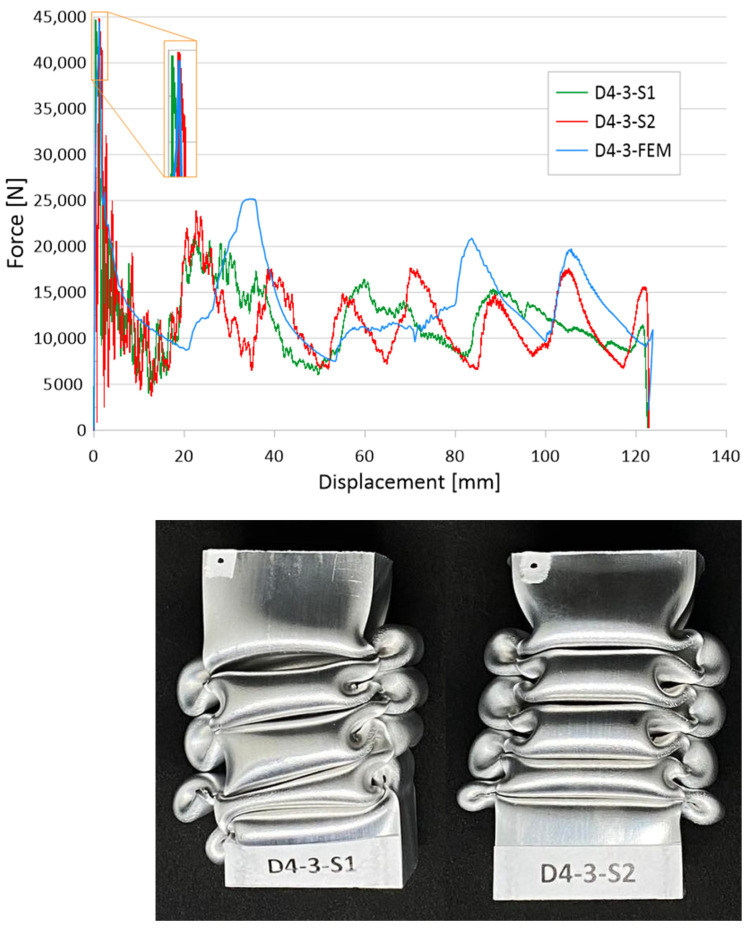
Comparison of numerical and experimental analysis based on D4-3 models.

**Table 1 materials-16-04196-t001:** Technical parameters of aluminum thin-walled tubes. (Reprinted with permission from Ref. [40]. Copyright 2023 Elsevier).

Name	AA-6063-T6							
Chemical composition (%)	Si	Fe		Cu	Mn	Mg	Zn	Ti
	0.54	0.23		0.008	0.051	0.481	0.005	0.021
Heat Treatment	8 h at 198 °C							
Mechanical property *	Re (MPa)		Rm (MPa)		A50 (%)		Brinell Hardness	
	240		265		11.5		80	

* According to EN ISO 6892-1.

**Table 2 materials-16-04196-t002:** Material properties of the aluminum alloy (our own research). (Reprinted with permission from Ref. [40]. Copyright 2023 Elsevier)

AA-6063-T6	
**Density ρ (kg/m^3^)**	2700
**Young’s Modulus E (MPa)**	70,000
**Poisson’s Ratio (–)**	0.33
**Stress σ (MPa)**	**Strain** **Ɛ (–)**
200	0
249.35	0.00248
279.98	0.0598

**Table 3 materials-16-04196-t003:** Sensitive analysis of the finite element method.

Size (mm)	No. of Elements (–)	Time (min.)	Discrepancy
1	31,945	180	107%
1.5	14,274	63	103%
2	8224	19	99%
2.5	5010	8	99%
3	3644	5	95%
3.5	2644	4	90%
4	2080	3	88%

## Data Availability

Data sharing not applicable.

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
