# Peer review of "Crashworthiness Analysis of Thin-Walled Square Columns with a Hole Trigger"

_materials, 2023, doi:10.3390/ma16114196_

Round 1

Reviewer 1 Report

The reviewer thinks that the author has done a good investigation on the research direction of this paper and has done a lot of work on the research of this paper, but there are the following questions in this paper:

1、In the first sentence of the third paragraph of Title 1, the article state that " researchers began to use structural reinforcement. "However, the follow sentence says "Due to the high stiffness in the longitudinal direction of the above structures, they have been replaced by porous structures used inside the column.", which is contrary to the previous statement.

2、It seems that there is a mistake in grammar in the second sentence of the fourth paragraph of Title1, that is, "An interesting take on composite metal-ceramic foam based on aluminum and silicon carbide (SiC) particles."

3、In Title 3.2, the author obtains the force-shortening characteristic with a reference point in the lower base. The reviewer suggests collecting force values in the higher base at the same time to make a comparison between them, although the influence of the stress waves is minimal in the case of low-speed impact.

4、In the last sentence of the second paragraph of Title4, some data of smooth column (numerical or experimental) are required to support the statement " the impact is evident compared to a smooth column without an initiator."

5、In the fourth sentence to the last of the fourth paragraph of Title 4, it is stated that "the two cases in question with the smallest hole diameter absorbed energy along the largest path". However, the Fig.6 displays that the displacement has a larger value when diameter is 2mm than that of 1.5mm, which is inconsistent with the explanation given by the author.

6、In the last two sentences of the fifth paragraph, the author state that "The value is much higher when referring to a smooth column without a crush initiator. Such a structure, through much higher stiffness in the direction of impact... " However, the stiffness should be reduced with holes in the sidewalls. Besides, there are not any evidence to prove the author’s point of view.

7、The reviewer suggests a brief explanation of "biomechanical properties" in the last paragraph of Title 4.

8、Are there any rules for determining the research parameters in experimental verification? Please explain why absorber with one initiation hole is not discussed in the experiment.

9、It is suggested to supplement the mesh convergence analysis in the simulation, and to calculate the influence of the mesh size on the calculation accuracy.

10、         Some relevant works are encouraged to mention. A novel equivalent method for crashworthiness analysis of composite tubes.

The reviewer thinks that the author has done a good investigation on the research direction of this paper and has done a lot of work on the research of this paper, but there are the following questions in this paper:

1、In the first sentence of the third paragraph of Title 1, the article state that " researchers began to use structural reinforcement. "However, the follow sentence says "Due to the high stiffness in the longitudinal direction of the above structures, they have been replaced by porous structures used inside the column.", which is contrary to the previous statement.

2、It seems that there is a mistake in grammar in the second sentence of the fourth paragraph of Title1, that is, "An interesting take on composite metal-ceramic foam based on aluminum and silicon carbide (SiC) particles."

3、In Title 3.2, the author obtains the force-shortening characteristic with a reference point in the lower base. The reviewer suggests collecting force values in the higher base at the same time to make a comparison between them, although the influence of the stress waves is minimal in the case of low-speed impact.

4、In the last sentence of the second paragraph of Title4, some data of smooth column (numerical or experimental) are required to support the statement " the impact is evident compared to a smooth column without an initiator."

5、In the fourth sentence to the last of the fourth paragraph of Title 4, it is stated that "the two cases in question with the smallest hole diameter absorbed energy along the largest path". However, the Fig.6 displays that the displacement has a larger value when diameter is 2mm than that of 1.5mm, which is inconsistent with the explanation given by the author.

6、In the last two sentences of the fifth paragraph, the author state that "The value is much higher when referring to a smooth column without a crush initiator. Such a structure, through much higher stiffness in the direction of impact... " However, the stiffness should be reduced with holes in the sidewalls. Besides, there are not any evidence to prove the author’s point of view.

7、The reviewer suggests a brief explanation of "biomechanical properties" in the last paragraph of Title 4.

8、Are there any rules for determining the research parameters in experimental verification? Please explain why absorber with one initiation hole is not discussed in the experiment.

9、It is suggested to supplement the mesh convergence analysis in the simulation, and to calculate the influence of the mesh size on the calculation accuracy.

10、         Some relevant works are encouraged to mention. A novel equivalent method for crashworthiness analysis of composite tubes.

Author Response

Responses to all reviewer questions have been included in the word file. All changes implemented in the manuscript are marked in yellow.

Reviewer 2 Report

Dear authors, you cand find my report in the attached file.

Kind regards.

Author Response

(The authors gave the same response as above.)

Round 2

Reviewer 2 Report

Dear authors.

Thank you so much for your kind and polite response.

I believe that the work has increase its appearance in this current version. In my opinion, it has been a good choice to include the values from the smooth specimen (free of holes) in figs. 7 and 8.

I still think that it is difficult to see the data evolution of figs. 4-6 with clarity, but I understand your explanation too.

Finally, to be honest, I still think that the key conclusions of the work (point 5) are independent of the Finite Element Analysis showed in the work. The key conclusions of the work are obtained from experimental results, not from numerical results (In my opinion the numerical results showed doesn’t contribute to clarify the key questions). Maybe I am wrong, but this is my feeling.

Nonetheless, I believe that your work is very nice and interesting, like I wrote in my previous report, and I think that this second version is better that the previous one. Congratulations!

Kind regards.